



# Distinct impacts on precipitation by aerosol radiative effect over three different megacity regions of eastern China

Yue Sun[1], Chuanfeng Zhao[1]

[1]College of Global Change and Earth System Science, and State Key Laboratory of Earth Surface Processes and Resource Ecology, Beijing Normal University, Beijing 100875, China

*Correspondence to:* Chuanfeng Zhao (czhao@bnu.edu.cn)

**Abstract.** Many studies have investigated the impacts of aerosol on the intensity and amount of precipitation, but few have been done regarding the impacts of aerosol on the start and peak time of precipitation. Using the high-resolution precipitation, aerosol and meteorological data in warm season of June-August from 2015 to 2020, this study investigates the influence of aerosol on the start and peak time of precipitation over three different regions, the North China Plain (NCP), the Yangtze River Delta (YRD), and the Pearl River Delta (PRD). It shows that the period with the most occurrence frequency of precipitation start time, defined as the frequent period (FP) of precipitation start time, is delayed and prolonged by aerosols in NCP, contributing to the similar durations of precipitation in NCP, YRD, and PRD. This study also shows that different types of aerosol (absorbing versus scattering) have caused different influences on the start and peak time of precipitation over the three study regions. The precipitation start time is 3 hours advanced in NCP but 2 hours delayed in PRD by aerosols during precipitation FP, and shows no response to aerosol in YRD. Compared to stratiform precipitation, the convective precipitation is more sensitive to aerosol. The start and peak time of convective precipitation show similar response to aerosols. This study further shows that the aerosol impact on precipitation can vary with meteorological conditions. Humidity is beneficial to precipitation, which can advance the precipitation start and peak time and prolong the precipitation duration time. Correspondingly, the impacts of aerosol on start time of precipitation are significant under low humidity or weak low tropospheric stability condition. The impacts of vertical wind shear (WS) on the start and peak time of precipitation are contrary to that of aerosols, resulting in the fact that WS inhibits the aerosol effects on precipitation.



## 1. Introduction

Aerosols can modify radiative energy balance, cloud physics, and precipitation and then affect both weather and climate, bringing large uncertainties to weather forecast and climate assessment (Edenhofer and Seyboth, 2013; Tao et al., 2012). Associated with the rapid economic development in China, heavy aerosol pollution has also resulted in serious impacts on atmospheric environment, weather, climate, and even public health (An et al., 2019; Song et al., 2017; Wang et al., 2017). Although the $PM_{2.5}$ mass concentrations have decreased significantly since 2013 due to the major air pollution control measures made by Chinese government (Ding et al., 2019; Fan et al., 2020; Wang et al., 2020; Zhang et al., 2020; Zheng et al., 2018), China is still among the regions with high aerosol amount. Thus, it is still necessary to further investigate the aerosol's impacts in China.

The aerosol can affect the cloud and precipitation by changing the radiation directly and by serving as cloud condensation nuclei (CCN) or ice nuclei (IN), which are referred as radiative effect and microphysical effect. On one hand, the aerosols can scatter and absorb solar radiation, which can heat the atmosphere and cool the surface, stabilize the atmosphere, and then suppress precipitation. Particularly, aerosol by absorbing solar radiation can strengthen the evaporation of cloud and then suppresses the formation of cloud and precipitation (Ackerman et al., 2000). On the other hand, aerosols, by serving as CCN or IN, can increase cloud droplet number concentration, resulting in larger cloud albedo (Twomey, 1977), enhanced cloud thermal emissivity (Garrett and Zhao, 2006; Zhao and Garrett, 2015), reduced precipitation and longer cloud lifetime (Albrecht, 1989; Pincus and Baker, 1994), and invigored convective precipitation (Fan et al., 2015; Li et al., 2011; Rosenfeld et al., 2008).

The aerosols show distinct influences on precipitation under different climatic regions, which make humid areas wetter and arid areas drier (Huang et al., 2006a; Huang et al., 2006b; Huang et al., 2010; Koren et al., 2005; Rosenfeld, 2000; Teller and Levin, 2006; Wang, 2005). Using long-term ground site observations, Li et al. (2011) have found that the increasing aerosols make the cloud higher and deeper under humid condition, which can increase the frequency and intensity of precipitation significantly and then increase the probability of floods; while under dry condition, aerosols can inhibit the development of cloud and precipitation and then increase the probability of drought. Based on the global satellite data, Niu and Li (2012) have further found that the above phenomenon is shown not only at single ground site, but even more pronounced in tropical regions. Considering the complexity of



precipitation processes and their variations with locations, studying the aerosol-precipitation
interactions is important to improve the accuracy of regional weather forecasts (Fan et al., 2015).
The significant influences of aerosol on cloud and precipitation in China have been reported in many
studies. In the southeast China, with the increase of the aerosol, the light and moderate precipitations
are inhibited, while the heavy precipitations are enhanced (Shi et al., 2015; Wu et al., 2015; Yang et al.,
2018). The aerosols over urban region can increase the total amount of precipitation in the case with
sufficient moisture supply and decrease the total precipitation amount in the case with insufficient
moisture supply (Chen et al., 2015; Qiu et al., 2017). Yang et al. (2017) found that the aerosols can
reduce the precipitation areas and intensity over Beijing-Tianjin-Hebei region using WRF-Chem model
simulations. Zhao et al. (2018) indicated that the aerosols can reduce the precipitation intensity while
enlarge the precipitation area of tropical cyclones over western pacific area using long-term
observations.
Most existing studies about the impacts of aerosol on precipitation have focused on the precipitation
amount, frequency, and intensity, but few studies have investigated how the aerosols affect
precipitation time, including both start and peak time of precipitation. Several studies have pointed out
that aerosols can make cloud higher and deeper under polluted condition, which will delay the
precipitation and cause strong thunderstorm precipitation in downwind areas (Andreae et al., 2004; Lin
et al., 2006; Rosenfeld et al., 2008). However, this effect, called as invigoration effect, has not gained
widely recognition. Several model simulation studies have shown that the invigoration effect is weak
and the aerosols even suppress convection in case with strong wind shear or with clod cloud base (Fan
et al., 2013; Fan et al., 2012; Fan et al., 2009; Khain et al., 2005; Lebo and Morrison, 2014). Moreover,
the delay caused by the invigoration effect has not yet been quantified.
The limited studies regarding the influence of aerosol on precipitation time showed controversial
findings in China. Yang et al. (2017) found that aerosols show no influence on precipitation time in
Beijing-Tianjin-Hebei region using WRF-Chem model simulations, while Zhou et al. (2020) reported
that aerosols advance the heavy precipitation start and peak time significantly, and prolong the duration
of the precipitation in Beijing-Tianjin-Hebei (BTH) region. Similar researches have been carried out by
Guo et al. (2016) and Lee et al. (2016) in Pearl River Delta (PRD) region. Guo et al. (2016) found that
the aerosol can delay heavy precipitation, which was further confirmed by model simulations (Lee et
al., 2016). Guo et al. (2016) and Lee et al. (2016) found that the aerosol radiative effect is dominant in



the initial stage of convection and the microphysical effect is dominant in the development stage, and
the interaction of radiative and microphysical effects eventually delays precipitation.
The controversial findings from limited previous studies raise a serious question: Why do the aerosols
show different impacts on the start and peak time of precipitation over different regions? To answer
this question, this study investigates the impacts of aerosols on the start and peak time of precipitation
over three different regions of North China Plain (NCP), YRD, and PRD by using data from the same
source with the same analysis method. With the support of high-precision data, this study tries to
quantify the impacts of aerosols on precipitation time. The responses of convective and stratiform
precipitation to aerosols are also investigated based on the precipitation type. Moreover, the changes of
aerosol impacts on precipitation time with meteorological conditions that can affect precipitation have
also been investigated, including the relative humidity, low troposphere stability, and vertical wind
shear, which are essential to aerosol-cloud-precipitation interactions (Boucher and Quaas, 2012; Fan et
al., 2009; Klein, 1997; Slingo, 1987; Zhou et al., 2020).
The paper is organized as follows. Section 2 describes the data and methods used in this study. Section
3 shows the analysis and results. The summary and discussion are provided in section 4.
**2. Data and methods**
**2.1 Region of Interest**
Three study regions of NCP, YRD, and PRD have been selected in this study, where the concentration
and types of aerosols are different. The $PM_{2.5}$ mass concentration decreases gradually from north to
south in China. The mixed-absorbing aerosols are dominant in NCP, which can absorb solar radiation
strongly and then heat atmosphere, followed by urban and industrial aerosols (Chen et al., 2014; He et
al., 2020). The dominant aerosols in the YRD are urban, industrial and mixing-absorbing aerosols (Che
et al., 2018; Chen et al., 2013; Chen et al., 2014; He et al., 2020). The main aerosol types in the PRD
are urban and industrial aerosols (Chen et al., 2014; He et al., 2020). It is worth noting that the
absorbing aerosols increase in North China Plain and Yangtze River Delta in June and August due to
biomass burning (Che et al., 2018; Chen et al., 2014).
Figure 1 shows the study region with surface altitude (m) information from Digital Elevation Model
(DEM), along with the location of $PM_{2.5}$ ground site stations. Due to the topographic rain effect, this

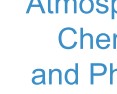
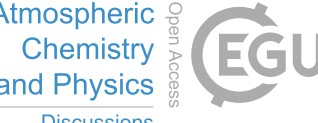

study only selects the area with DEM less than 100 meters as the study region. There are 131, 100, and
70 ground sites in NCP, YRD, and PRD, respectively. In order to obtain enough precipitation samples
and then reduce the statistical error, the selected study period is the summer (June to August) of
multiple years from 2015 to 2020.
**2.2 Data**
The datasets including precipitation, aerosol, and meteorological fields are used in this study, which
are described as follows.
**2.2.1 Precipitation data from GPM**
The Global Precipitation Measurement (GPM) mission can provide global observations of rain and
snow. Compared to the Tropical Rainfall Measuring Mission (TRMM), the GPM extends capability to
measure light rain (< 0.5 mm/hr), solid precipitation, and the microphysical properties of precipitating
particles, in addition to the ability of observing heavy to moderate precipitation. The observation
devices are the first space-borne Ku/Ka band Dual-frequency Precipitation Radar (DPR) and a
multi-channel GPM Microwave Imager (GMI). The DPR Level-2A product is used in this study.
The DPR instrument can provide three dimensional measurements of precipitation structure over 78
and 152 miles (125 and 245 km) swaths. The combination of detection information from the Ka band
precipitation radar (KaPR) and Ku band precipitation radar (KuPR) can retrieve precipitation particle
size distribution and snowfall events effectively, which is beneficial to facilitate the understanding of
precipitation nature and structure deeply. The DPR Level-2A product with a temporal resolution of 90
minutes provides precipitation profile data from ground to 21,875 meters at 125 meters vertical
intervals, including precipitation position, type, and intensity, the height of freezing level, the height of
storm top, and so on.
GPM generally performs better for summer, liquid precipitation, and plain area than for winter, solid
precipitation, and complex terrain area (Chen et al., 2019; Speirs et al., 2017). This study focuses on
the warm season in eastern China and the precipitation is mostly liquid during the study period, so the
DPR Level-2A product is suitable to be used. A major role of the DRP Level-2A product in this study is
to classify the three types of precipitation, which are convective, stratiform, and other.



### 2.2.2 Hourly precipitation from China Merged Precipitation Analysis Version 1.0 product

The other precipitation dataset used in this study is the hourly China Merged Precipitation Analysis Version 1.0 product. This product has a spatial resolution of 0.1° and a temporal resolution of 1 hr in China. The hourly precipitation product is downloaded online (ftp://nwpc.nmc.cn). The product is developed based on the observation data at 30,000 automatic stations in China and Climate Prediction Morphing Technique (CMORPH) data. This product overcomes the shortcoming from ground stations that is difficult to provide the change of the spatial distribution of the overall climate due to discontinuous distribution. Simultaneously, this product overcomes the issue of poor accuracy of satellite products. With these merits, this dataset has been successfully applied to many precipitation-related studies (Guo et al., 2016; Sun et al., 2019), which provides us the possibility for examining aerosol impacts on precipitation time in this study.

### 2.2.3 Aerosol data

This study takes use of the hourly $PM_{2.5}$ mass concentration provided by the China Environmental Monitoring Station of the national air quality real time release platform with data quality assurance (http://beijingair.sinaapp.com) to represent aerosol. Previous studies have used AOD or $PM_{10}$ to study the influence of aerosol on precipitation (Guo et al., 2016; Zhao et al., 2018; Zhou et al., 2020). However, AOD could be not suitable for many cases since it represents the column-integrated aerosol amount while precipitation mostly occurs in the troposphere and is more affected by aerosols below cloud bases. $PM_{10}$ might be also not suitable for the study of aerosol impacts on precipitation particularly in case large aerosol particles such as dust exist since $PM_{10}$ is more representative of large aerosol particles while cloud condensation nuclei is more related to the aerosol particle number with sizes larger than 100 nm. Instead, $PM_{2.5}$ mass concentration is more representative of aerosol particle amount with sizes larger than 100 nm, so that we choose $PM_{2.5}$ to represent the aerosol amount in this study.

The diurnal variation of $PM_{2.5}$ mass concentration is significant in the study regions, especially over NCP as shown later. This diurnal variation raises a question for the study of aerosol impacts on precipitation: what time should we choose for the aerosol observations that have more clear impacts on precipitation? Figure 2 shows the relationship of $PM_{2.5}$ mass concentration between the daily mean and the 7:00-12:00 LT mean, the 13:00-18:00 LT mean, the value in 1 hour before precipitation, the mean





value in 2 hours before precipitation, the mean value in 3 hours before precipitation, the mean value in
4 hours before precipitation, and the mean value in 5 hours before precipitation. As shown, the
correlation between daily mean $PM_{2.5}$ mass concentration and 7:00-12:00 LT (13:00-18:00 LT) mean
$PM_{2.5}$ mass concentration is relatively poor (r=0.57-0.73) in the three study regions. The correlation
coefficients between the daily mean $PM_{2.5}$ mass concentration and $PM_{2.5}$ mass concentration averaged
in 1 (2, 3, 4, 5) hours before precipitation are worse than that between daily mean $PM_{2.5}$ mass
concentration and 7:00-12:00 LT (13:00-18:00 LT) mean $PM_{2.5}$ mass concentration, suggesting that it is
not suitable to use $PM_{10}$ mass concentration or AOD at a given moment to examine the influence of
aerosol on precipitation. Taking into account that the aerosol effect needs time to accumulate, this study
selects the 4-hours mean $PM_{2.5}$ mass concentration before precipitation to investigate the impact of
aerosols on precipitation.
**2.2.4 ERA5**
As indicated earlier, three essential meteorological variables will be investigated in this study, which
are the relative humidity, low troposphere stability, and vertical wind shear. Relative humidity can
affect both precipitation process and AOD. And the clouds occurring is closely related to water vapor,
for example clear skies were more likely than cloudy skies for relative humidities below 65% (Boucher
and Quaas, 2012; Klein, 1997; Slingo, 1980, 1987; Zhou et al., 2020). The low troposphere stability
can signify the strength of the inversion that caps the planetary boundary layer, which is correlated with
cloud amount (Klein, 1997; Wood and Bretherton, 2006). High LTS generally means a relatively stable
atmospheric stratification and low LTS means unstable atmospheric column, which is more favorable
for the development of convection (Guo et al., 2016; Klein, 1997; Slingo, 1987). Wind shear implies
mechanical turbulence, which can influence detrainment and evaporation of cloud hydrometeors and
then affects the aerosol effect on precipitation (Fan et al., 2009; Slingo, 1987; Tao et al., 2007). Fan et
al. (2009) found that the vertical wind shear plays a dominant role in regulating aerosol effects on
isolated deep convective clouds, which determines whether aerosols suppress or enhance convection.
The meteorological datasets including the three key variables shown above are from ERA5 in this study,
which is the fifth generation ECMWF (European Centre for Medium-Range Weather Forecasts,
ECMWF) reanalysis data (https://cds.climate.copernicus.eu/). The ERA5 is better than the
ERA-Interim in temporal-spatial resolutions of 1 hour and 0.25°×0.25°, respectively, and have



contributed to thousands of studies (e.g., Fan et al., 2020; Hoffmann et al., 2019; Urraca et al., 2018;
Yang et al., 2021). The ERA5 hourly data on pressure levels are used in this study, including
temperature (at 1000, 975, 950, 925, 900, 875, and 850 hPa), relative humidity (at 850 hPa), vertical
velocity (at 1000, 975, 950, 925, 900, 875, and 850 hPa) and wind (at 850, and 500hPa) on different
pressure levels.

### 2.3 Methods

The hourly precipitation product is shown in grid pattern, but the $PM_{2.5}$ mass concentration dataset is
from site observation. Therefore, the matching between precipitation information and $PM_{2.5}$ mass
concentration is not point to point. However, the representative area of $PM_{2.5}$ site observation is
between 0.25 and 16.25 $km^2$ (Shi et al., 2018), and the representative area is even larger in clean and
plain areas, so the vague matching described as follows should be reasonable. Assuming the location of
$PM_{2.5}$ site is a given point called as A, and the point A is in a certain grid of hourly precipitation product
that is called as B, the $PM_{2.5}$ mass concentration at A can then be used to represent the pollution
condition at B. In order to know the precipitation type at B, we find the nearest location according to
the latitude and longitude provided by GPM. The ERA5 dataset is also shown in grid pattern and we
use the same method described above to match hourly precipitation product and the ERA5 dataset.
The main method used in this study is cluster analysis. We divide all study period into three groups
based on the $PM_{2.5}$ mass concentration, and defined two of them as polluted and clean conditions to
further investigate the aerosol impacts on precipitation. The detailed method is as follows. First, we sort
all observations of $PM_{2.5}$ by removing the abnormal values that are over 2 times the standard deviation
to get the good quality data group C. Second, we rank the $PM_{2.5}$ mass concentration observations from
high to low, and define the top 1/3 of group C as clean condition and the bottom 1/3 group C as
polluted condition. Similar classification method has been applied to other variables when defining
their high and low value conditions, such as meteorological conditions including the low troposphere
stability (LTS), vertical wind shear between 1500 m to 5500 m (WS), and relative humidity (RH). The
LTS (unit: K) used here is the difference of potential temperature at 700 hPa and 1000 hPa (Slingo,
1987; Wood and Bretherton, 2006). The relative humidity (unit: %) at 850 hPa is used to represent the
moisture below the cloud base in this study (Klein, 1997; Zhou et al., 2020). The wind shear (unit: $s^{-1}$)
can be calculated as (Guo et al., 2016),
$$WS = \frac{\sqrt{(u_{5.5}-u_{1.5})^2+(v_{5.5}-v_{1.5})^2}}{(5500-1500)} \ldots\ldots\ldots (1)$$

where $u_{5.5}$ and $u_{1.5}$ are horizontal wind speed at 5500 m and 1500 m, respectively; $v_{5.5}$ and $v_{1.5}$ are
vertical wind speed at 5500 m and 1500 m, respectively. The wind speed at 1500 (5500) m can be
converted to wind speed at 500 (850) hPa by barometric height formula.
**3. Results**
**3.1 Characteristics of PM$_{2.5}$ and precipitation**
Figure 3 shows the diurnal variation of PM$_{2.5}$ mass concentration. As shown, the diurnal variation of
PM$_{2.5}$ mass concentration is strong in NCP and weak in YRD and PRD, which further confirms that the
too long time average of PM$_{2.5}$ mass concentration cannot reliably represent the aerosol amount that
influence the precipitation during a relatively short term. The diurnal variation patterns of PM$_{2.5}$ are
similar in NCP, YRD, and PRD, with low values in the afternoon and high values at night, along with
high PM$_{2.5}$ mass concentration values in rush hours. The diurnal variations of PM$_{2.5}$ is most likely
related to the diurnal variation of boundary layer height (BLH). The high BLH is conducive to the
diffusion of pollutants in the afternoon, while the low BLH is not conducive to the diffusion at night.
Moreover, the PM$_{2.5}$ mass concentration is also high around 12:00 LT in PRD, which is most likely
caused by the secondary formation by strong solar radiation.
This study focuses on the start and peak time of precipitation event. We define the precipitation event
as a continuous precipitation, that is, no precipitation before and after this precipitation at least for 1
hour. During a precipitation event, the time that precipitation appears is called start time, and the time
that precipitation intensity is the highest is called peak time. Figure 4 shows the statistical probability
density function (PDF) of precipitation start and peak time. There are more than 800 samples at any
given hour in the study regions, make the results statistically convincing. As shown in Figure 4, the
precipitation events are more frequent at 14:00-16:00 LT but less frequent at 6:00-8:00 LT, which are
corresponding to the time of strong and weak solar radiation, respectively. In general, the cloud
droplets occur when the atmosphere gets saturated and the droplets can further become precipitation
particles through the processes of condensational growth, collision-coalescence, and so on. Strong solar
radiation can increase the atmospheric instability by heating the ground surface, further enhancing the
convection and promoting the formation of precipitation. In the following analysis, we set the



continuous periods that over the red dotted line as the period with most frequent occurrence of
precipitation (simply called Frequent Period) and we set the periods that below the red dotted line as
Infrequent Period. There are subtle differences in the Frequent Periods of the start time (shown in
Figure 4a, 4b, and 4c) and peak time (shown in Figure 4d, 4e, and 4f) of precipitation over the same
region. Note that we use Frequent (Infrequent) Period (S) and Frequent (Infrequent) period (P) to
denote the Frequent (Infrequent) Periods of start time and peak time, respectively.
As shown in Figure 4a-c, the Frequent Periods and Infrequent Periods are different significantly in the
three study regions. The Frequent Period (S) is 14:00-21:00 LT in NCP, 11:00-19:00 LT in YRD, and
11:00-18:00 in PRD. The durations of Frequent Period (S) are 8, 9, and 8 hours in NCP, YRD, and PRD,
respectively. The initial time of Frequent Period (S) in NCP is three hours later than that in YRD and
PRD, likely suggesting that the solar radiation takes longer time to strengthen convection in NCP than
in YRD and PRD. In contrast, the Frequent Periods (S) turn into Infrequent Periods (S) soon after
sunset in YRD and PRD, while the Frequent Period (S) remains 3 hours after sunset in NCP. This
makes the initial time of the Frequent Period (S) different but the durations similar in the three study
regions. It is curious why the Frequent Period (S) can remain 3 hours after sunset in NCP and what
powers the precipitation or convection during the 3 hours. Figure 3 already shows that the $PM_{2.5}$ mass
concentration is the highest in NCP and the lowest in PRD. In addition, there is a relatively large
proportion of aerosols as absorbing type in NCP comparing to that in YRD and PRD (Yang et al., 2016).
As known, the aerosol can heat the atmosphere and cool the ground by scattering and absorbing solar
radiation. Thus, it is most likely that the large quantities of aerosol particles in NCP weaken the
downward surface shortwave radiation in the morning and make the Frequent Period (S) delayed.
Simultaneously, the large quantities of aerosol particles could release the heat they absorbed in the low
atmosphere to extend the Frequent Period (S) of precipitation after sunset.
The diurnal variation of peak time of precipitation is similar to that of the start time, also with more
frequent occurrence in the afternoon and less frequent occurrence in the early morning. As shown in
Figure 4d-f, the Frequent Periods (P) are 14:00-21:00, 12:00-20:00, 11:00-19:00 LT in NCP, YRD, and
PRD, respectively, which indicates that the peak time is often 1-2 hours later than the start time. In NCP,
although the Frequent Period (S) and Frequent Period (P) are the same, the frequency of precipitation
peak time at 14:00 LT is lower than that for the precipitation start time, while the frequency at
15:00-16:00 LT is higher than that for the precipitation start time, which further confirms that the peak



time is often 1-2 hours later than the start time.
Figure 5 shows the PDFs of the precipitation duration time and when the peak time occurs after start
time. As shown, precipitation events within 2 hours account for more than 50% of all precipitation
events, and the precipitation events within 4 hours account for more than 80% of all precipitation
events. In fact, long-time precipitation events are mostly related to large-scale weather systems, and the
impact of aerosol on them is difficult to identify from the complex meteorological factors. Therefore,
the precipitation events selected in this study are those with duration time within 4 hours. As shown in
Figure 5d-e, because of the high proportion of short-term precipitation events, the peak time tends to
occur shortly after the precipitation start time. More than 90% of the precipitation peak time occur
within 4 hours of the precipitation events.
Table 1 shows the sample volume of precipitation events along with the precipitation types obtained
from GPM product. There are totally 21,567 matched precipitation events in NCP, with 78.60%
(16,951 cases) as stratiform precipitation and 15.59% (3,362 cases) as convective precipitation. The
number of other precipitation events is small, so this study does not investigate the other precipitation
further. The numbers of precipitation events are 30,659 and 26,861 in YRD and PRD, respectively. The
proportions of stratiform precipitation events are higher than 56% both in YRD and PRD, and the
proportion of convective precipitation is secondary to the stratiform precipitation with values more than
21%. As shown in Table 1, the proportions of convective precipitation gradually increase and the
proportions of stratiform precipitation gradually decrease from NCP, YRD to PRD.
**3.2 Influence of aerosol on precipitation start (peak) time**
We investigate the influence of aerosol on precipitation start and peak time by analyzing their Frequent
Period and Infrequent Period, respectively. Figure 6 shows the PDFs of the start and peak time of
precipitation events under polluted and clean conditions. During the Frequent Period of precipitation in
NCP, the crest of start time is 15:00 LT under polluted condition and 18:00 LT under clean condition,
which implies that the start time of precipitation is 3 hours advanced by aerosols. In the Infrequent
Period of precipitation start time in NCP, the influences of aerosol on the start time of precipitation are
different between before and after sunrise: the start time is 1-2 hours delayed by aerosol after sunrise
while there is no significant delay or advance in start time of precipitation by aerosol before sunrise.
The diurnal variations of precipitation start time are similar in pattern between polluted and clean





conditions in YRD, suggesting that aerosols have no significant impact on the precipitation start time
over YRD. In addition, the crest of precipitation start time during the Frequent Period is about 16:00 LT
under both clean and polluted in YRD. Figure 4 already shows that the crest of precipitation start time
is at 14:00 LT in PRD. Figure 6c further shows that the crest of precipitation start time is at 13:00 LT
under clean condition and at 15:00 LT under polluted condition in PRD during the Frequent Period of
precipitation, while there are no obvious differences in the PDFs of precipitation start time between
polluted and clean conditions during the Infrequent Period.
Above results shown in Figure 6 clearly suggest that the influences of aerosol on the start time of
precipitation are distinct over the three study regions, especially during their Frequent Period. The
aerosol can advance, delay, or show almost no effect on the crest of the start time over the NCP, PRD,
and YRD, respectively. Moreover, the aerosols make precipitation more focused in the afternoon and
suppress the precipitation at night over all three study regions, which is most obvious over PRD. The
diurnal variations of the precipitation start time are much more different between the polluted and clean
conditions in PRD. During the period 12:00-22:00 LT, the frequency of precipitation under polluted
condition is higher than that under clean condition, while during the other period contrary phenomenon
is found in PRD.
We also investigate the influence of aerosol on the precipitation peak time during their Frequent Period.
The diurnal variations and the responses of precipitation peak time to aerosol are similar to that of the
precipitation start time. By comparing the diurnal variations of precipitation peak time under polluted
and clean conditions, we find that although the aerosols can advance or delay the precipitation time, the
diurnal variation pattern has not been changed. Based on the almost fixed patterns, we can quantify the
impacts of aerosol on the precipitation start and peak time. As shown earlier, we can investigate the
crest of the precipitation start and peak time to quantify the influence of aerosol on the precipitation,
but this method is not always suitable. As shown in Figure 6d, the crests of the peak time are at 15:00
and 18:00 LT under polluted and clean conditions during the Frequent Period respectively, which
suggests that the aerosol has caused the precipitation peak time 3 hours advanced in NCP. However, by
comparing the diurnal variations of precipitation peak time between polluted and clean conditions, we
find that there are secondary crests of precipitation peak time at 17:00 and 16:00 LT under the polluted
and clean conditions respectively, which suggests that the aerosol has caused the precipitation peak
time 1 hour advanced. Anyway, what we can confirm from Figure 6d is that the high frequency of the


precipitation peak time is at 15:00-17:00 LT under polluted condition while at 16:00-18:00 LT under
clean condition. During the Infrequent Period over NCP, there are relatively more precipitation under
polluted condition than under clean condition before sunrise, while there are relatively less
precipitation under polluted condition after sunrise. Also, the precipitation peak time is 1 hour delayed
(advanced) over NCP under polluted condition after (before) sunrise during the Infrequent Period of
precipitation.
The crests of the precipitation peak time are both at 16:00 LT under polluted and clean conditions over
YRD during the Frequent Period, which suggests that the aerosols show negligible impact on the
precipitation peak time. In contrast, it shows that the precipitation peak time is 1 hour advanced under
polluted condition during the Infrequent Period over YRD. The diurnal variations of the precipitation
peak time are similar to that of the precipitation start time both under polluted and clean conditions
over PRD. The precipitation peak time over PRD has been 2 hours delayed during the Frequent Period
and 1 hour advanced during the Infrequent Period (before sunrise) by aerosols. The responses of
precipitation start and peak time to aerosol are similar with each other. Consistent with the fact that the
precipitation peak time appears 1-2 hours after the precipitation start time as shown in Figure 5, the
crest of the precipitation peak time is also later than that of the precipitation start time as shown in
Figure 6.
The findings above show that the aerosols have distinct impacts on the precipitation start time in NCP
(advanced), YRD (no influence), and PRD (delayed), which may be related to their different aerosol
amount and types, precipitation types, or meteorological conditions. Among the three study regions, the
most polluted area is NCP and the cleanest area is PRD. Meanwhile, the proportion of the absorbing
aerosol is the highest in NCP and is the lowest in PRD. Both aerosol concentration and the proportion
of the absorbing aerosol in YRD are between NCP and PRD, based on which the mechanism that
aerosol impacts the precipitation over YRD should include that over both NCP and PRD if the aerosols
do have significant impacts on precipitation. The initial time of the Frequent Period in NCP (14:00 LT)
is later than that in PRD (11:00 LT), which is most likely due to the high aerosol concentration in NCP.
The high aerosol concentration reduces the solar radiation reaching the ground, making the convection
suppressed in the morning in NCP. However, the high proportion of absorbing aerosol can advance the
precipitation start time by strengthening the convection in the afternoon. In contrast, the scattering
dominant aerosol can cool the ground surface and then low atmosphere by scattering solar radiation,





which weakens the convection and generally delays the precipitation start time during the Frequent
Period in PRD. We also find that the aerosol makes the precipitation more frequent at night in NCP,
which is most likely associated with the fact that the aerosol can heat the atmosphere and strengthen
convection even after sunset due to the relatively high proportion of absorbing aerosol in NCP. In
addition to aerosols, we also find that the variation of meteorology can play a role to the change of
precipitation. For example, the decreasing temperature and increasing humidity are both contributable
to the growth of cloud droplets and then precipitation at night. After sunrise, the precipitation seems
more influenced by solar radiation and aerosols in NCP. The atmosphere is heated more quickly under
clean condition than under polluted condition in the morning in NCP, making the probability of
precipitation higher under clean condition in the morning.
The precipitation is also affected by solar radiation and aerosols after sunrise in YRD, but the aerosols
show no significant influence on the precipitation start time likely due to weak radiative effect by the
relatively low aerosol amount over this study region. Even with weak radiative effect due to relatively
low aerosol amount, the aerosol still makes the precipitation more frequent in the afternoon and more
infrequent in the morning and at night over YRD, which likely suggests the significant aerosol
microphysical effect on the precipitation. Aerosols, by serving as cloud condensation nuclei, increase
the cloud droplet number concentration and decrease cloud droplet sizes, decreasing the stratiform
precipitation that occurs more in the morning and invigorating the convective precipitation that occurs
more in the afternoon.
To further understand whether the different precipitation types cause distinct responses of precipitation
to aerosols, we next investigate the impacts of aerosol on convective and stratiform precipitation using
the same method. Note that we ignore some hours in a day, at which the sample size is too small (less
than 10) to be analyzed reliably and we only investigate the impacts of aerosol on convective and
stratiform precipitation during the continuous period of precipitation.
Figure 7 shows the PDFs of convective (stratiform) precipitation start time under polluted (red line)
and clean (blue line) conditions. Figs. 7a-c show that the convective precipitation occurs frequently at
time around 8:00, 12:00-14:00, and around 18:00-20:00 LT, and infrequent at 15:00-16:00 LT and at
night in NCP. The aerosols advance convective precipitation start time 1-2 hours during 10:00-15:00 LT,
while show no obvious influence during the periods 0:00-9:00 LT and 16:00-20:00 LT in NCP.
Consistent with the results presented above, aerosol makes the precipitation more accumulated in the



afternoon, particularly at days when the aerosol radiative effect works strongly. The convective
precipitations are found frequently at 9:00-15:00 LT in YRD. The crest of convective precipitation start
time is both at 12:00 LT under polluted and clean conditions during the period 8:00-16:00 LT in YRD,
while it is delayed by 1 hour by aerosols during the period 13:00-16:00 LT. The continuous period with
enough precipitation samples is 7:00-22:00 LT in PRD. The convective precipitation start time over
PRD shows negligible response to aerosols during the period 7:00-11:00 LT, while is 1 hour delayed
during the period 12:00-22:00 LT. As shown in Figure 7c, the crest and secondary crest of the
convective precipitation start time are at 12:00 and 17:00 LT under clean condition and at 14:00 and
18:00 LT under polluted condition, which implies that the delaying effect of aerosols on convective
precipitation start time becomes weaker with the decreasing solar radiation or convective strength.
Figs. 7d-f show the stratiform precipitation occurs frequently at night and around sunrise with a peak
occurrence frequency at about 7:00 LT in NCP. The aerosol shows no significant influence on the start
time of the stratiform precipitation in NCP. In YRD, the diurnal variations of the stratiform
precipitation start time are similar under polluted and clean conditions, while the occurrence
frequencies at a given hour are slightly different, which indicates that the aerosol can only weakly
affect the stratiform precipitation start time. In PRD, more stratiform precipitation occurs in the
afternoon under polluted condition. Moreover, the crests of the stratiform precipitation start time are at
19:00 and 17:00 LT under clean and polluted conditions in the afternoon, respectively, which suggests
that the aerosol could advance the stratiform precipitation start time by 2 hours in PRD.
Figure 8 shows the PDFs of the convective and stratiform precipitation peak time under polluted and
clean conditions. Note that only the continuous periods with >10 precipitation events at each given
hour are investigated. The continuous periods with convective precipitation are 0:00-15:00 LT and
17:00-22:00 LT in NCP. As shown in Figure 8a, the crests of the convective precipitation peak time are
at 13:00 LT (polluted condition) and 15:00 LT (clean condition) in NCP, which suggests that the aerosol
could advance the convective precipitation peak time by 2 hours during the period 0:00-15:00 LT.
However, it is challenging to identify whether the convective precipitation peak time has been changed
by aerosols during the period 17:00-22:00 LT because of the discontinuous distribution of convective
precipitation in NCP. The convective precipitations are frequent during the period 10:00-17:00 LT and
aerosols show no significant influence on the convective precipitation peak time in YRD. For example,
the crests of convective precipitation peak time are both at 14:00 LT under clean and polluted



conditions during the period 10:00-17:00 LT, one of the continuous periods with sufficient samples of
convective precipitation events in YRD. Figure 8c shows that there is a continuous period of
convective precipitation at 0:00-17:00 LT in PRD, during which the aerosol enhances the convective
precipitation gradually. The radiative effect of aerosol generally works significantly during the period
11:00-15:00 LT, which helps advance the convective precipitation peak time by 1 hour in PRD.
The frequency of the stratiform precipitation of the day fluctuates greatly in NCP, and shows larger
values in the early morning and early afternoon over YRD. The stratiform precipitations are not
affected by aerosols clearly over both NCP and YRD. Over PRD, the stratiform precipitation is also
strengthened gradually by aerosol, while the stratiform precipitation peak time is likely 1 hour delayed
by aerosols during the period 13:00-21:00 LT. It is clear that the aerosol affects the convective
precipitation much more strongly than the stratiform precipitation over NCP and YRD, while the
aerosol shows different impacts on convective and stratiform precipitation over PRD. Due to the high
proportion of the stratiform precipitation over PRD, the start and peak time of total precipitation events
are delayed, as shown in Figure 6.
The above findings have suggested that the aerosol can affect convection, and we next try to confirm
this hypothesis. If the aerosol could affect precipitation and convection, the temperature and vertical
velocity would show strong responses to the changes of aerosol over the plain regions. We here
investigate how the temperature and vertical velocity change with aerosol concentration and type at
different pressure levels. The differences of temperature between polluted and clean conditions are
shown in Figure 9a-c. As shown, the aerosol causes significant changes of atmospheric temperature by
radiative effect at low troposphere (1000-900 hPa). As the altitude increases, the aerosol radiative effect
decreases gradually which results in smaller temperature differences. The strongest influence of aerosol
on temperature is shown in NCP and the weakest is in PRD, which is likely related to their difference
in aerosol amount. It is also clear that the aerosol heats the atmosphere all day in NCP.
As shown in Figure 9a, the radiative effect of aerosol is strengthened gradually after the sunrise with
the largest impact on atmospheric temperature at 19:00-22:00 LT and gets weakened from midnight to
before sunrise the next day in NCP, which implies that the precipitations are also affected by the
aerosol radiative effect at night. The atmosphere is heated by aerosols over YRD for almost all time
except the period 3:00-6:00 LT. The radiative effect of aerosol increases after sunrise and decreases
after sunset with the largest impact on atmospheric temperature at 15:00-18:00 LT in YRD. The





obvious cooling effect of aerosol is shown in PRD for almost all time except for a weak heating effect
in the morning. After sunrise, the cooling effect increases gradually in PRD. The above phenomena
could help explain why the aerosol shows different influence on the precipitation start and peak time
over the three study regions. Over the NCP, the impacts of aerosol radiative effect on atmospheric
temperature at 1000-950 hPa is weaker than that at 925-875 hPa, implying that the potential convective
energy need time to accumulate. Correspondingly, the convection is strengthened weakly in the
morning even though the aerosol can heat the atmosphere due to the high aerosol concentration.
Accompanied by the accumulation of aerosol heating effect with time, the aerosols favor the
convection strongly and then advance the precipitation start time over the NCP. Differently, the
aerosols paly a cooling effect over the PRD, and accompanied by the accumulated aerosol cooling
effect with time, the precipitation start time is delayed.
Figure 9d-f show the differences in vertical velocity between polluted and clean conditions, which
further confirms the above results. The positive vertical velocity (downward movement) suppresses the
convection and the negative (upward movement) strengthens the convection. In general, when the
aerosol heats (cools) the atmosphere, the airflow is updraft (downdraft). However, we should note
when the radiative effect of aerosol is weak (at night and in the early morning), the increasing
temperature does not mean that the airflow must be updraft.
**3.3 Sensitivities of aerosol impacts on precipitation to meteorological factors**
In addition to aerosols, meteorological variables can also affect the precipitation. We here investigate
the potential impacts from the meteorological variables, and further investigate the aerosol impacts on
precipitation by limiting the influence from those meteorological variables. This study selects three
crucial factors for the precipitation formation and development, including moisture, wind shear and low
troposphere stability (Fan et al., 2009; Guo et al., 2016; Klein, 1997; Slingo, 1987; Zhou et al., 2020).
Figures S1-S3 show the influence of moisture, WS and LTS on precipitation. Sufficient moisture is
beneficial to precipitation generation and advances precipitation. The differences in precipitation
frequency between crest and valley under high humidity condition are less than that under low
humidity condition, which means that high moisture increases the precipitation frequency for all
corresponding time instead of making precipitation gathered at a particular time range. As a result, the
high humidity weakens the diurnal variations of precipitation frequency. The LTS changes the diurnal




characteristics of the precipitation start time. The precipitation is more frequent in the daytime with
peak occurrence frequency in the afternoon under low LTS condition, while the precipitation is more
frequent at the nighttime with valley occurrence frequency in the afternoon under high LTS condition.
The high WS delays the precipitation start time by 3 hours in NCP, delays the precipitation start time
by 1 hour in YRD, and advances the precipitation start time by 2 hours in PRD, which is opposite to the
influence of aerosol on precipitation start time. Therefore, the high WS inhibits the aerosol effects on
precipitation, which is in good agreement with the findings by Fan et al. (2009) that increasing aerosol
concentrations can enhance convection under weak wind shear condition.
Using the similar method to classify meteorological conditions as aerosols, this study next investigates
the differences of crest or valley of precipitation frequency between polluted and clean conditions to
verify the aerosol effects by limiting the meteorological conditions. Under high humidity condition, the
diurnal variations of precipitation frequency are more complicated under polluted condition over the
NCP and YRD, making it challenging to judge the corresponding crest and valley time. Moreover, the
aerosol radiative effect is weak under high humidity condition, which could also make the impacts of
aerosols on precipitation hard to identify. Under low humidity condition, the aerosols advance the
precipitation start time by 3 hours in NCP and by 1 hour in YRD. The aerosols delay the precipitation
start time by 2 hours both under low and high humidity conditions in PRD. However, the differences of
PDFs between polluted and clean conditions under low humidity condition are more distinct than that
under high humidity condition over the PRD, which indicates that the aerosol effects on precipitation
are more significant under low humidity condition. All above results suggest that the humidity can
affect the strength of aerosol impacts on precipitation. The aerosol impacts on precipitation are more
obvious under low humidity condition and are somehow weakened under high humidity condition. The
response of aerosol impacts on precipitation peak time to humidity is basically consistent with that of
the aerosol impacts on precipitation start time, but shows weakened aerosol impacts under high
humidity condition more clearly, especially in PRD. Under low humidity condition, the crest of
precipitation peak time is at 14:00 LT under clean condition and at 16:00 LT under polluted condition,
suggesting that the precipitation peak time is 2 hours delayed by aerosols in PRD. Differently, under
high humidity condition, the crests of precipitation peak time are both at 15:00 LT under both polluted
and clean conditions, which suggests that the aerosols have no obvious influence on precipitation peak
time under high humidity condition in PRD.



Figure 11 shows that the aerosol effects on precipitation are distinct under low LTS condition but are
almost negligible under high LTS condition. The aerosols make the precipitation start time in NCP and
YRD 1 hour advanced under low LTS condition. During the Frequent Period of precipitation, the
frequency of precipitation under polluted condition is higher than that under clean condition, which
means that the aerosol microphysical effect is prominent in addition to the aerosol radiative effect. The
precipitation start time is 2 hours delayed (polluted: 16:00 LT, clean: 14:00 LT) by aerosol in PRD. The
response of precipitation peak time to the aerosols are generally consistent with that of precipitation
start time under different LTS conditions. The aerosol impacts on precipitation are distinct under high
and low WS conditions while they are more obvious under low WS condition. In the NCP, the aerosols
advance the precipitation start time under both low and high WS conditions, which suggests that the
aerosol radiative effect plays significant role. However, under low WS condition, the crest frequency of
precipitation under polluted condition is higher than that under clean condition in NCP, while contrary
phenomenon is found under high WS condition, which suggests that the high WS suppresses the
aerosol microphysical effects. The aerosols make the precipitation start time 1 hour earlier under low
WS condition in YRD while the aerosol effects on precipitation start time are not obvious under high
WS condition. The aerosols delay the precipitation start time under both low and high WS conditions in
PRD. The responses of precipitation peak time to aerosols are also found generally consistent with that
of precipitation start time under different WS conditions.
**4. Summary and discussion**
**4.1 Summary**
This study investigates the influence of aerosol on the precipitation start and peak time over three
different megacity regions using the high-resolution precipitation, aerosol, and meteorological datum in
summer (June-August) during the period from 2015 to 2020. We first examine the changes of
precipitation start and peak time with aerosols over the North China Plain (NCP), the Yangtze River
Delta (YRD), and Pearl River Delta (PRD) regions. Then we classify the precipitation into convective
and stratiform precipitation types, and examine their different responses in start and peak time to
aerosols. Finally, considering that meteorological variables, particularly three key meteorological
variables of humidity, low tropospheric stability, and wind shear, also play important roles to



precipitation development, we further classify the meteorological conditions using the same method as
aerosols and examine the aerosol impacts on precipitation start and peak time under different
meteorological conditions. New findings have been provided with the following several key points.
1) The Frequent Period of precipitation start time is delayed and prolonged by high aerosol
concentrations and relatively high proportion of absorbing aerosol in NCP, so the initial time of the
Frequent Period in NCP (14:00 LT) is later than that in YRD (11:00 LT) and PRD (11:00 LT) while the
durations of Frequent Periods are similar among the three study regions. The different aerosol
concentrations and aerosol types (absorbing versus scattering) contribute to the different aerosol
impacts on the precipitation start (peak) time over the NCP, YRD and PRD. The precipitation start time
is 3 hours advanced in NCP but 2 hours delayed in PRD by aerosols during the Frequent Period and the
precipitation start time in YRD shows negligible response to aerosol. The most likely reason is that the
aerosol heats the atmosphere strongly in NCP, associated with the high aerosol concentration and the
relatively larger proportion of absorbing aerosol over the NCP. The aerosol concentration and aerosol
type in PRD is opposite to that in NCP. The aerosol concentration and aerosol type in YRD both are
between that in NCP and PRD, and the aerosol impacts on the precipitation start (peak) time in YRD
are also between that in NCP and PRD, which is relatively weakly affected by aerosol. The influences
of aerosol radiative effect on precipitation start (peak) time are also found different during the different
periods of the day.
2) The frequency of stratiform precipitation is higher than that of convective precipitation, but the
convective precipitation is more sensitive to aerosol than stratiform precipitation. The responses of the
convective precipitation start and peak time to aerosol are similar to each other with the results as
shown above in point 1), except that the start time is 1 hour delayed in YRD, but the peak time is 1
hour advanced in PRD.
3) Humidity is beneficial to precipitation which can advance the precipitation start (peak) time, but the
influence of aerosol on precipitation is weakened when the humidity is high. The low tropospheric
stability (LTS) can modify the diurnal variation characteristics of precipitation start (peak) time. The
influences of aerosol on precipitation start time are more significant under low LTS. Vertical wind
shear (WS) inhibits the aerosol effects on precipitation, since the influences of WS on the precipitation
start (peak) time are opposite to that of aerosols. WS delays the precipitation start (peak) time by 3
hours in NCP and by 1 hour in YRD, while advances the precipitation start (peak) time by 2 hours in

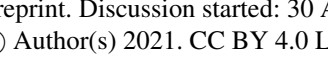



PRD.
**4.2 Discussion**
The aerosol-precipitation interaction is a hot topic in atmospheric science and has many challenges due
to its complexity. Previous studies have focused on the influence of aerosols on the precipitation
intensity at inter-decadal or daily time scales, but few studies have examined the impacts of aerosols on
the precipitation time for a large amount of precipitation events. This study investigates the impacts of
aerosols on the precipitation start and peak time for both stratiform and convective precipitations by
limiting the impacts of meteorological variables, which are essential for improve our understanding of
aerosol-precipitation interaction. However, there are still some problems in current study, with at least
the following several points.
First, the temporal resolution of observations is still too coarse for current study. For example, the
temporal resolution of precipitation product is 1 hour in this study, which makes it difficult for us to
more accurately quantify the impact of aerosols on precipitation time: precipitation time changes with
values less than 1 hour are not able to be identified. Second, the complicated mechanisms and
processes of aerosol effect on precipitation could introduce extra uncertainties to our findings.
Currently, we only examine the sensitives of aerosol effects on precipitation under different humidity,
LTS and WS conditions, which might be not sufficient. Also, this study focuses on summer
precipitation, but the influence of summer monsoon has not been considered and definitely need be
investigated further in future. Finally, we would like to mention that we focus on the aerosol radiative
effects on precipitation time while the aerosol microphysical effect is less discussed. It is hard to
distinguish radiative effect and microphysical effect using observation study alone, so the numerical
model simulations should be applied further in future. Moreover, the influence of aerosol on
precipitation intensity and duration also need to be investigated deeply further over different regions.
***Data availability.*** Surface elevation data from the Shuttle Radar Topography Mission (SRTM) were
downloaded from http://srtm.csi.cgiar.org/ (Yang et al., 2021). ERA-5 Reanalysis data were provided
by the European Centre for Medium Weather Forecasts (https://cds.climate.copernicus.eu/, Fan et al.,
2020). The hourly precipitation data from China Merged Precipitation Analysis Version 1.0 product can
be downloaded in real time from the http://cdc.cma.gov.cn/sksj.do?_method=ssrjscprh (Shen et al.,





2014). The hourly PM$_{2.5}$ mass concentration provided by the China Environmental Monitoring Station
of the national air quality real time release platform with data quality assurance
(http://beijingair.sinaapp.com, Sun et al., 2019). The DPR Level-2A product from the Global
Precipitation Measurement (GPM) mission can be downloaded from
https://gpm.nasa.gov/missions/GPM (Zhang et al., 2018).
*Author contributions.* CFZ and YS developed the ideas and designed the study. YS contributed to
collection and analyses of data. YS performed the analysis and prepared the manuscript. CFZ
supervised and modified the manuscript. All authors made substantial contributions to this work.
*Competing interests.* The authors declare that they have no conflict of interest.
*Acknowledgements.* This work was supported by the Ministry of Science and Technology of China
National Key Research and Development Program (2017YFC1501403), the National Natural Science
Foundation of China (41925022, 41575143), the State Key Laboratory of Earth Surface Processes and
Resources Ecology, and the Fundamental Research Funds for the Central Universities.

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

**Figures and tables**
Table 1: The number and proportion of different types of precipitation in the three study regions of
North China Plain (NCP), Yangtze River Delta (YRD), and Pearl River Delta (PRD).

| Study area | NCP | YRD | PRD |
|---|---|---|---|
| Total case numbers | 21567 | 30659 | 26861 |
| Convective case numbers (proportion %) | 3362 (15.59) | 6683 (21.8) | 9464 (35.23) |
| Stratiform case numbers (proportion %) | 16951 (78.6) | 21104 (68.83) | 15309 (56.99) |
| Other case numbers (proportion %) | 1254 (5.81) | 2872 (9.37) | 2088 (7.77) |

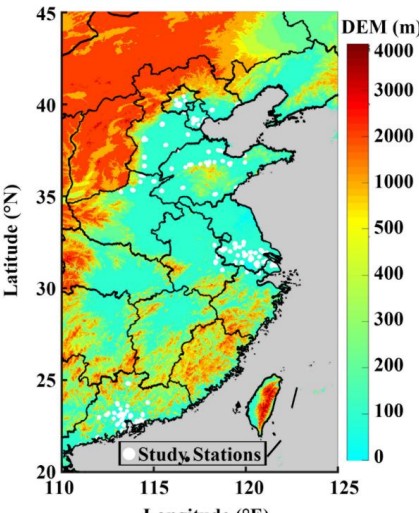


Figure 1: The study region with surface altitude (m) information from Digital Elevation Model (DEM).
The white dots are the $PM_{2.5}$ site stations used in this study, and the color map represents the DEM
information.

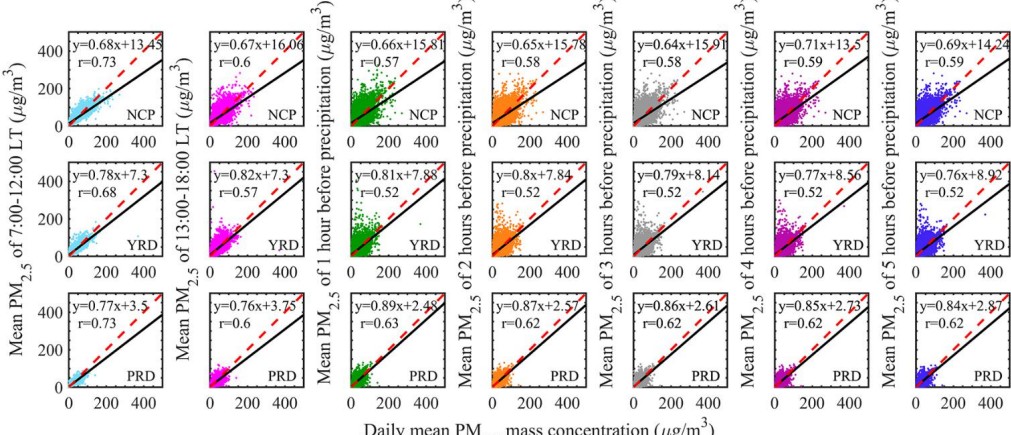


Figure 2: The relationships between the daily mean PM$_{2.5}$ mass concentration (μg/m$^3$) and the mean

PM$_{2.5}$ mass concentration of 7:00-12:00 LT (azure, the first column), 13:00-18:00 LT (roseo, the second

column), 1 hour before precipitation (green, the third column), 2 hours before precipitation (orange, the

fourth column), 3 hours before precipitation (grey, the fifth column), 4 hours before precipitation

(purple, the sixth column), and 5 hours before precipitation (blue, the seventh column) in June-August

from 2015 to 2020 over North China Plain (NCP, the first row), Yangtze River Delta (YRD, the second

row), and Pearl River Delta (PRD, the third row), respectively.

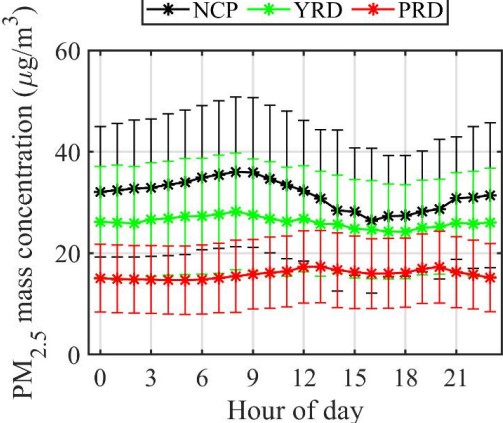


Figure 3: The diurnal variation of PM$_{2.5}$ mass concentration (μg/m$^3$) during the period of June-August

from 2015 to 2020 in North China Plain (NCP; black), Yangtze River Delta (YRD; green) and Pearl





River Delta (PRD, red). The dotted lines are for average values, and the vertical bars are for standard
deviations of PM$_{2.5}$ mass concentration at each hour.

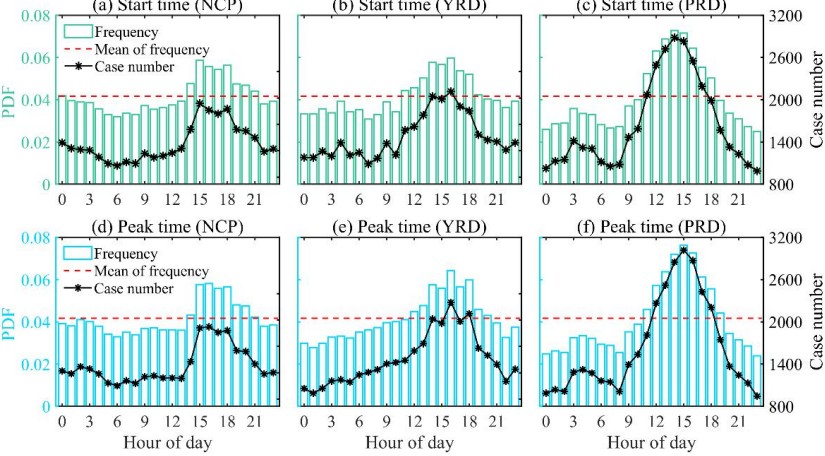


Figure 4: The probability density functions (PDFs) of the start time (a-c, green) of precipitation and the
peak time (d-f, blue) of precipitation in June-August from 2015 to 2020 over three study regions. The
NCP, YRD, and PRD represent North China Plain, Yangtze River Delta, and Pearl River Delta,
respectively. The black line represents the sample amount of precipitation events at the corresponding
time, and the red dotted line is the average daily precipitation frequency.

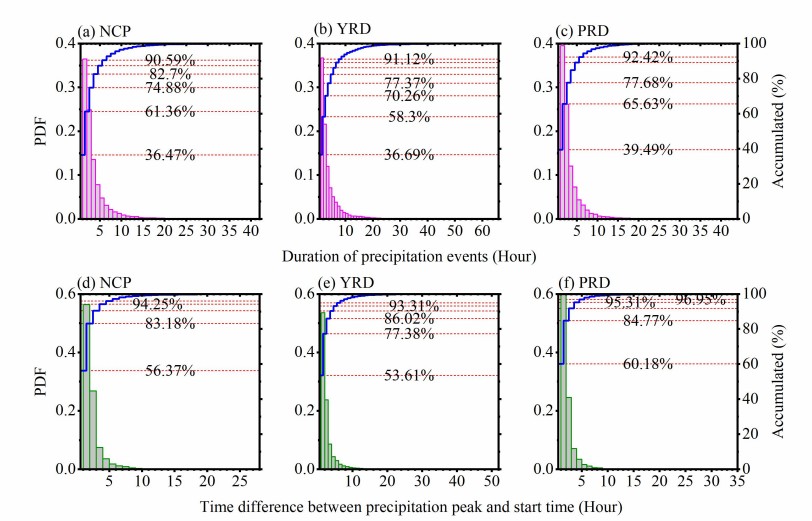


Figure 5: The PDFs of duration of precipitation events (a-c) and PDFs of time difference (in hours)
between precipitation peak and start time for all precipitation events (d-e) during the study period of
June-August from 2015 to 2020 over three study regions. The NCP, YRD, and PRD represent North
China Plain, Yangtze River Delta, and Pearl River Delta, respectively. Blue solid lines denote
accumulated occurrence frequencies of precipitation (ordinate on the right-hand side of each panel).
Red dotted lines and numbers show the accumulated occurrence frequencies of precipitation.

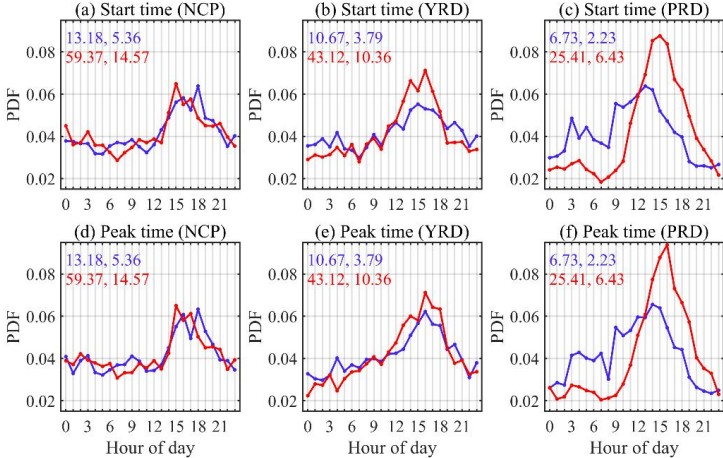


Figure 6: Normalized PDFs of precipitation (a-c) start time and (d-e) peak time (units: LT), represented
as ratios of their corresponding precipitation frequency at a given hour to those accumulated over 24 h
under clean (blue lines) and polluted (red lines) conditions in June-August from 2015 to 2020 over
NCP, YRD and PRD, respectively. The blue (red) numbers are the average (the first column) and
standard deviation (the second column) of the $PM_{2.5}$ mass concentration ($\mu g/m^3$) under clean (polluted)
condition.

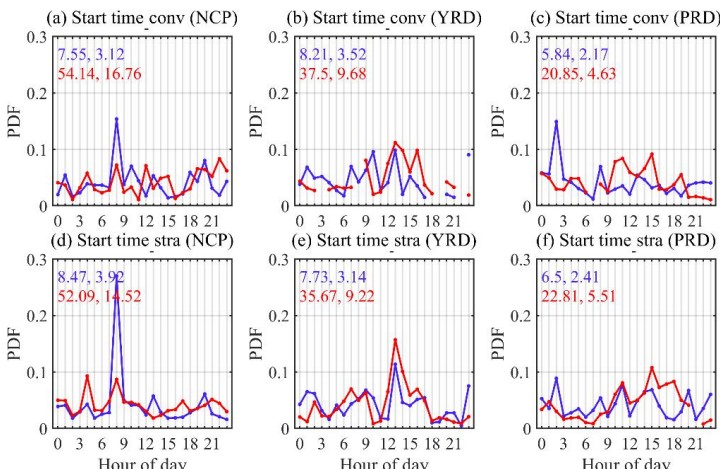

Figure 7: Normalized PDFs of (a-c) convective precipitation start time and (d-e) stratiform precipitation start time (units: LT), represented as ratios of their corresponding precipitation frequency at a given hour to those accumulated over 24 h under clean (blue lines) and polluted (red lines) conditions in June-August from 2015 to 2020 over NCP, YRD and PRD, respectively. The blue (red) numbers are the average (the first column) and standard deviation (the second column) of the $PM_{2.5}$ mass concentration ($\mu g/m^3$) under clean (polluted) condition.

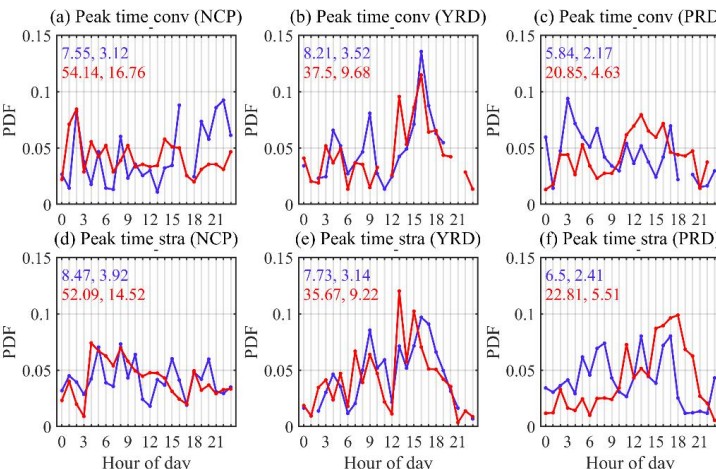

Figure 8: Same as Figure 7, but for (a-c) convective precipitation peak time and (d-e) stratiform precipitation peak time (units: LT).



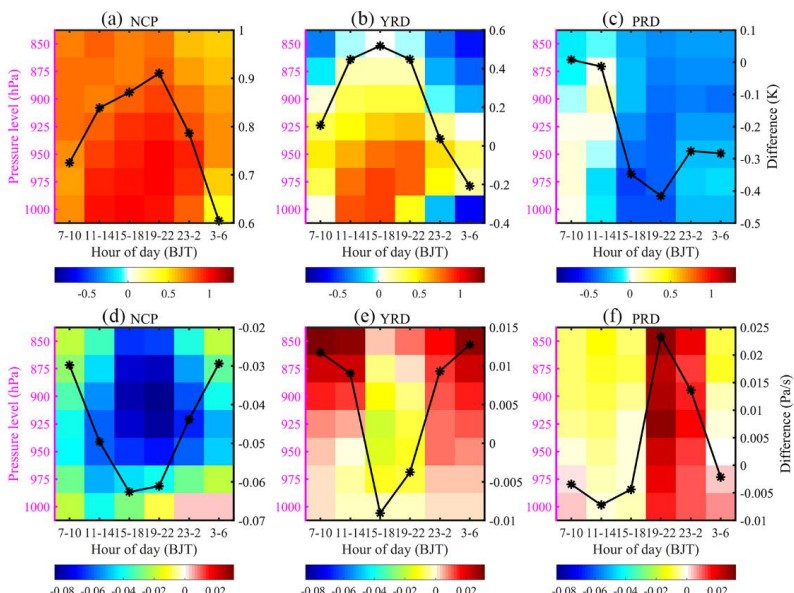

875

Figure 9: The differences in (a-c) temperature (K) and (d-f) vertical velocity (Pa/s) between polluted

and clean conditions in NCP, YRD and PRD at different pressure levels. The positive (negative) values

represent heating (cooling) of the atmosphere in (a-c). The positive (negative) values represent down

(up) airflow in (d-f). The black lines represent the means of the differences in temperature (vertical

velocity) from 1000 to 850 hPa for several given hour periods, including 7:00-10:00, 11:00-14:00,

15:00-18:00, 19:00-22:00, 23:00-2:00 (the next day) and 3:00-6:00 LT.

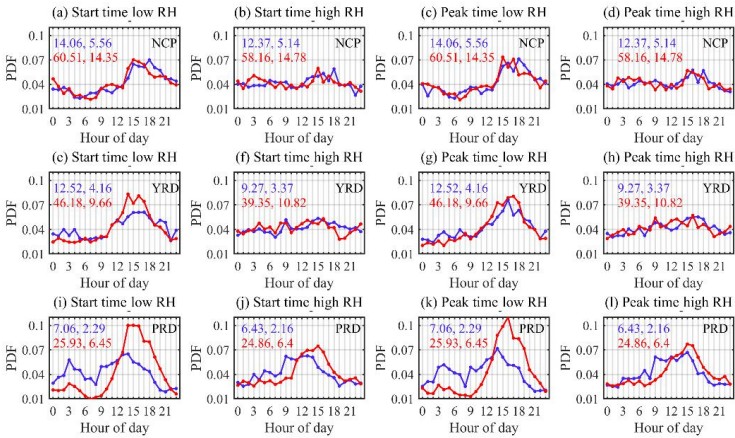






Figure 10: Normalized PDFs of precipitation start time under (a, c, i) low humidity condition and (b, f,
j) high humidity condition, the precipitation peak time under (c, g, k) low humidity condition and (d, h,
l) high humidity condition in June-August from 2015 to 2020 over NCP, YRD and PRD, respectively.
The blue (red) numbers are the average (the first column) and standard deviation (the second column)
of the PM$_{2.5}$ mass concentration (μg/m$^3$) under clean (polluted) condition. The RH represents the
relative humidity.

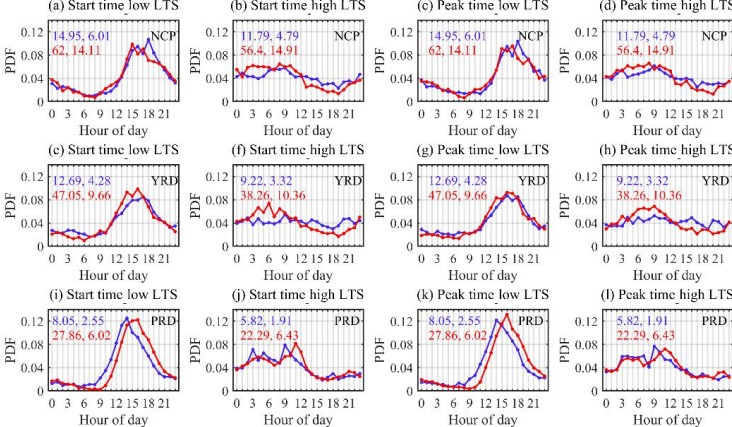


Figure 11: Same as Figure 10, but under low LTS condition and high LTS condition. The LTS
represents low troposphere stability.

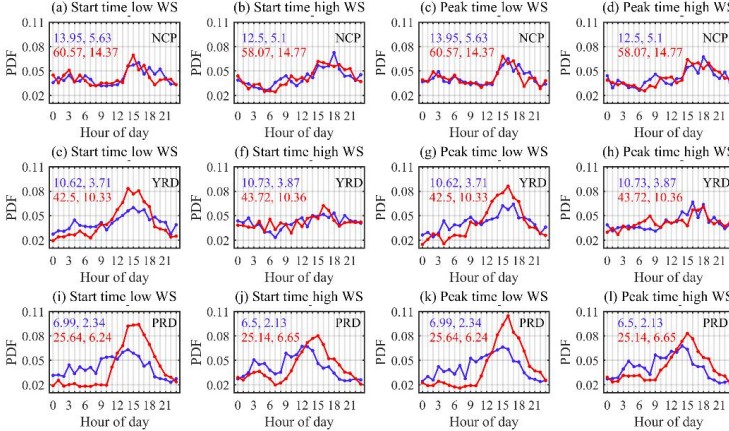


Figure 12: Same as Figure 10, but under low WS condition and high WS condition. The WS represents
vertical wind shear between heights at 5500 m and 1500 m.