# Peer review of "Distinct impacts on precipitation by aerosol radiative effect over three different megacity regions of eastern China"

_Atmospheric Chemistry and Physics, 2021_

## Author Comment (AC1)

Replies to Referee #1's comments

We thank the reviewer for the thoughtful, valuable and detailed comments and suggestions that have helped us improve the paper. Our detailed responses (Blue) to the reviewer's questions and comments (Italic) are listed below.

*The manuscript "Distinct impacts on precipitation by aerosol radiative effect over three different megacity regions of eastern China" mainly studies the influence of aerosol on the start and peak time of precipitation over three different regions, the North China Plain (NCP), the Yangtze River Delta (YRD), and the Pearl River Delta (PRD). In general, the paper is well written and presented in a logical way. It is a timely and important piece of work, and of general interest for Atmospheric Chemistry and Physics related communities. I therefore recommend publication of this paper in Atmospheric Chemistry and Physics after minor revisions. My comments are listed as follows:*

We highly appreciate the reviewer's positive evaluation about our study and have made corresponding changes based on the reviewer's comments.

Specific Comments:

*Lines 158-160: If precipitation occurs in the troposphere and is more affected by aerosols below cloud bases, why is the column-integrated aerosol amount (AOD) not suitable but ground-based observations of $PM_{2.5}$ are more suitable?*

This is a good question. AOD could be not suitable for many cases since it represents the column-integrated aerosol amount while precipitation mostly occurs in the troposphere and is more affected by aerosols below cloud bases. Besides, the AOD is not a good proxy for CCN (Stier, 2016) and is strongly correlated to humidity (Boucher and Quaas, 2012). While ground-based aerosol observations are also not the aerosols at cloud bases, most convective clouds investigated here with precipitation are with cloud bases near the tops of mixed boundary layer (MBL). Considering that aerosols are generally well mixed within the MBL layer, the ground-based observations are more suitable for this study. Comparing to $PM_{10}$, fine aerosols can serve as the best proxy for CCN comparing to coarse aerosols (Pan et al., 2021). Thus, we think that ground-based observations of $PM_{2.5}$ are more suitable. We have made corresponding changes in the manuscript at Lines 193-205.

Boucher, O. and Quaas, J.: Water vapour affects both rain and aerosol optical depth, Nat. Geosci., 6, 4-5, https://doi.org/10.1038/ngeo1692, 2012.

Pan, Z., Rosenfeld, D., Zhu, Y., Mao, F., Gong, W., Zang, L., and Lu, X.: Observational quantification of aerosol invigoration for deep convective cloud lifecycle properties based on geostationary satellite, J. Geophys. Res. Atmos., 126, e2020JD034275, https://doi.org/10.1029/202 0JD03 4275, 2021.

*Stier, P.: Limitations of passive remote sensing to constrain global cloud condensation nuclei, Atmos. Chem. Phys., 16, 6595–6607, https://doi.org/10.5194/acp-16-6595-2016, 2016.*

*Lines 160-165: Perhaps the authors' opinion is that $PM_{10}$ is more suitable for studying larger particle aerosols such as dust, and there are fewer large particle aerosols in the three selected research areas, so $PM_5$ is more suitable than $PM_{10}$ in this study. A clearer description is needed here. And what does "100 nm" represent?*

We appreciate the valuable information from the reviewer. What we would like to emphasize is that $PM_{10}$ is more representative of aerosol mass of large particles (particularly dust) while $PM_{2.5}$ is more representative of aerosol number concentration with sizes larger than 100 nm. Thus, $PM_{2.5}$ is more suitable to represent CCN than $PM_{10}$. We now added more description to clarify our selection at Lines 193-205:
"**Besides, the AOD is not a good proxy for CCN (Chen et al., 2021; Stier, 2016) and is strongly correlated to humidity (Boucher and Quaas, 2012).** $PM_{10}$ might be also not suitable for the study of aerosol impacts on precipitation particularly in case large aerosol particles such as dust exist since $PM_{10}$ is more representative of large aerosol particle mass while cloud condensation nuclei is more related to the aerosol particle number with sizes larger than 100 nm. **Pan et al. (2021) have reported that fine aerosols can serve as the best proxy for CCN comparing to AOD and coarse aerosols. Instead, $PM_{2.5}$ mass concentration is more representative of aerosol particle amount with sizes larger than 100 nm, so that we choose $PM_{2.5}$ to represent the aerosol amount in this study. Of course, there are few large particle aerosols in the three selected research areas (Fan et al., 2021), especially in summer. Also noted is that while the ground-based aerosol observations are not the aerosols at cloud bases, most convective clouds investigated here with precipitation are with cloud bases near the tops of mixed boundary layer (MBL). Considering that aerosols are generally well mixed within the MBL layer, the ground-based $PM_{2.5}$ is suitable to represent the aerosol amount below cloud bases in this study**".

*Boucher, O. and Quaas, J.: Water vapour affects both rain and aerosol optical depth, Nat. Geosci., 6, 4-5, https://doi.org/10.1038/ngeo1692, 2012.*

*Chen, T., Li, Z., Kahn, R. A., Zhao, C., Rosenfeld, D., Guo, J., Han, W., and Chen, D.: Potential impact of aerosols on convective clouds revealed by Himawari-8 observations over different terrain types in eastern China, Atmos. Chem. Phys., 21, 6199–6220, https://doi.org/10.5194/acp-21-6199-2021, 2021.*

*Fan, H., Zhao, C., Yang, Y. and Yang, X.: Spatio-Temporal Variations of the $PM_{2.5}/PM_{10}$ Ratios and Its Application to Air Pollution Type Classification in China, Front. Environ. Sci., 9:692440, http://doi.org.10.3389/fenvs.2021.692440, 2021.*

Pan, Z., Rosenfeld, D., Zhu, Y., Mao, F., Gong, W., Zang, L., and Lu, X.: *Observational quantification of aerosol invigoration for deep convective cloud lifecycle properties based on geostationary satellite, J. Geophys. Res. Atmos., 126, e2020JD034275,* https://doi.org/10.1029/202 0JD03 4275*, 2021.*

Stier, P.: *Limitations of passive remote sensing to constrain global cloud condensation nuclei, Atmos. Chem. Phys., 16, 6595–6607,* https://doi.org/10.5194/acp-16-6595-2016*, 2016.*

*Lines 177-179: Why can the previous phenomenon suggest that "it is not suitable to use $PM_{10}$ mass concentration or AOD at a given moment to examine the influence of aerosol on precipitation"?*

Sorry for our confusing description here. We have modified our description here at Lines 217-218: "……suggesting that **it is not suitable to use $PM_{2.5}$ mass concentration at a given moment to examine the influence of aerosol on precipitation.**". Before we propose this point, we have checked relationship of $PM_{2.5}$ mass concentration between the daily mean and the 7:00-12:00 LT mean, the 13:00-18:00 LT mean, the value at 1 (2, 3, 4, 5) hours before precipitation and found that the correlations among them are poor in the three study regions (shown in figure 2). Besides, $PM_{2.5}$ mass concentration shows obvious diurnal variations (shown in figure 3), so we think that it is not suitable to use $PM_{2.5}$ mass concentration at a given moment to examine the influence of aerosol on precipitation. Moreover, we should note that the aerosol radiative effect on convective clouds and precipitation needs time to accumulate.

*Line 180: Why do the authors select the 4-hours mean $PM_5$ mass concentration before precipitation to investigate the impact of aerosols on precipitation? The relationship between daily mean $PM_{2.5}$ and 4-hours mean $PM_{2.5}$ mass concentration before precipitation, similar to Figure 1, is needed.*

This is a good question. The relationship between daily mean $PM_{2.5}$ and 4-hours mean $PM_{2.5}$ mass concentration before precipitation have been shown in Figure 2 (purple, the sixth column). We select 4-hours mean $PM_{2.5}$ mass concentration before precipitation to investigate the impact of aerosols on precipitation based on the following two considerations. Firstly, $PM_{2.5}$ show strong diurnal variation. As shown in figure 2, the correlation between daily mean $PM_{2.5}$ mass concentration and 7:00-12:00 LT (13:00-18:00 LT) mean $PM_{2.5}$ mass concentration is relatively poor (r=0.57-0.73) in the three study regions. The correlation coefficients between the daily mean $PM_{2.5}$ mass concentration and $PM_{2.5}$ mass concentration averaged in 1 (2, 3, 4, 5) hours before precipitation are worse than that between daily mean $PM_{2.5}$ mass concentration and 7:00-12:00 LT (13:00-18:00 LT) mean $PM_{2.5}$ mass concentration. Therefore, it is not suitable to use daily mean $PM_{2.5}$ mass concentration or $PM_{2.5}$ at a given moment to examine the influence of aerosol on precipitation. Secondly, the aerosol effect needs time to accumulate, so this study selects the 4-hours mean $PM_{2.5}$

mass concentration before precipitation to investigate the impact of aerosols on precipitation.

*Line 189: What is the full name of LTS?*

Thanks for the question. The full name of LTS is low troposphere stability. We now add the full name and abbreviation of LTS when we first refer, which is at Lines 100-103: "Moreover, the changes of aerosol impacts on precipitation time with meteorological conditions that can affect precipitation have also been investigated, including the relative humidity, low troposphere stability **(LTS)**, and vertical wind shear **(WS)**, which are essential to aerosol-cloud-precipitation interactions (Boucher and Quaas, 2012; Fan et al., 2009; Klein, 1997; Slingo, 1987; Zhou et al., 2020).".

*Line 219: The authors regard $PM_{2.5}$ that is greater than 2 times the standard deviation as abnormal values and remove it, which could lead to mistakenly remove some heavy pollution conditions as abnormal values. Is it reasonable? And what is the proportion of the sample size that is eliminated as abnormal values in the total sample?*

Thank you for the question. We study the impact of aerosol on precipitation based on the large number of samples, the abnormal values among which can affect the characteristics of $PM_{2.5}$ mass concentration, especially the extreme abnormal values. (shown in Figure R1 d-f). Therefore, it is important to remove the abnormal values to clarify the subject features of $PM_{2.5}$. The proportions of abnormal values are 4%, 4%, and 5% in North China Plain, Yangtze River Delta, and Pearl River Delta, respectively. Previous studies removed some abnormal values that greater than 3 or 1 times the standard deviation (e.g. Fan et al., 2020) to investigate the variation of $PM_{2.5}$ mass concentration. As shown in Figure R1, the histograms of hourly $PM_{2.5}$ mass concentration that are smaller than 3, 2, and 1 time(s) the standard deviation are similar. The similar average values and standard deviation values under clean and polluted conditions among hourly $PM_{2.5}$ mass concentrations that are smaller than 3, 2, and 1 time(s) suggest different datum of $PM_{2.5}$ mass concentrations show negligible effect on the responses of precipitation to aerosol after removing abnormal values. Therefore, it is reliable to regard $PM_{2.5}$ that is greater than 2 times the standard deviation as abnormal values.

[Figure]

Figure R1: The histogram of hourly PM$_{2.5}$ mass concentration (μg/m$^3$) in June-August from 2015 to 2020 over three study regions. The NCP, YRD, and PRD represent North China Plain, Yangtze River Delta, and Pearl River Delta, respectively. The blue (brown, green) histogram represents hourly PM$_{2.5}$ mass concentration that is smaller than 3 (2, 1) times the standard deviation. The blue (brown, green) dotted line represents the threshold of clean condition and the solid line represents the threshold of polluted condition. The blue (brown, green) numbers are the average (the first and third column) and standard deviation (the second and fourth column) of the PM$_{2.5}$ mass concentration (μg/m$^3$) under clean and polluted condition.

*Fan, H., Zhao, C., and Yang, Y.: A comprehensive analysis of the spatio-temporal variation of urban air pollution in China during 2014-2018, Atmospheric Environ., 220, 117066.1-117066.12, https://doi.org/10.1016/j.atmosenv.2019.117066, 2020.*

*Lines 344-355: I think it suggests that the aerosol has caused the secondary crest of precipitation peak time 1 hour delayed.*

We are sorry for the confusing description. As shown in Figure 6d, the crests of the peak time are at 15:00 and 18:00 LT under polluted and clean conditions during the Frequent Period respectively, which suggests that the aerosol has caused the precipitation peak time 3 hours advanced in NCP. However, by comparing the diurnal variations of precipitation peak time between polluted and clean conditions, the right correspondence should be 15:00-16:00-17:00 LT and 16:00-17:00-18:00 LT under polluted and clean conditions, which suggests that the aerosol has caused the precipitation peak time 1 hour advanced instead of 3 hours advanced. We have corrected our descriptions at Lines 391-395: "**However, by comparing the diurnal variations of precipitation peak time between polluted and clean conditions, the right correspondence should be 15:00-16:00-17:00 LT and 16:00-17:00-18:00 LT**

**under polluted and clean conditions, which suggests that the aerosol has caused the precipitation peak time 1 hour advanced not 3 hours.**".

*Lines 422-424: The information of "the crests of the stratiform precipitation start time are at 19:00 and 17:00 LT under clean and polluted conditions in the afternoon, respectively" cannot be gotten from Figure 7f. It likely to be 20:00 and 18:00 LT under clean and polluted conditions, respectively.*

We highly appreciate the reviewer for helping figure out the mistake we made. We have corrected them at Lines 475-477: "Moreover, the crests of the stratiform precipitation start time are **at 20:00 and 18:00 LT** under clean and polluted conditions in the afternoon, respectively, which suggests that the aerosol could advance the stratiform precipitation start time by 2 hours in PRD.".

*Line 429: I think they are 14:00 and 16:00 in NCP from Figure 8a.*

We highly appreciate the reviewer for helping figure out this issue and we have made changes at Lines 480-483: "**The continuous periods with convective precipitation are 0:00-16:00 LT and 18:00-22:00 LT in NCP. As shown in Figure 8a, the crests of the convective precipitation peak time are at 14:00 LT (polluted condition) and 16:00 LT (clean condition) in NCP**, which suggests that the aerosol could advance the convective precipitation peak time by 2 hours during the period 0:00-16:00 LT.".

*Lines 527-529: Please clearly indicate which figure and which situation are aimed at.*

We are sorry for the unclear description and have added related information, which are at Lines 578-583: "Under low humidity condition, the crest of precipitation peak time is at 14:00 LT under clean condition and at 16:00 LT under polluted condition, suggesting that the precipitation peak time is 2 hours delayed by aerosols in PRD **(Figure 10k)**. Differently, under high humidity condition, the crests of precipitation peak time are both at 15:00 LT under both polluted and clean conditions **(Figure 10l)**, which suggests that the aerosols have no obvious influence on precipitation peak time under high humidity condition in PRD.".

Following the reviewer's suggestion, we have also added corresponding information to indicate which figure and which situation are aimed at at other lines in the manuscript.

---

## Author Comment (AC2)

**Replies to Referee #2's comments**

We thank for your thoughtful, valuable and detailed comments and suggestions that have helped us improve the paper. Our detailed responses (Blue) to the reviewer's questions and comments (Italic) are listed below.

This manuscript reports on the distinct impacts on precipitation start/peak time by aerosol radiative effect over three different megacity regions of eastern China, which is found mainly caused by the different aerosol concentration and types over the three regions. The manuscript argues that the precipitation start time is 3 hours advanced in North China Plain due to high proportion of absorbing aerosol, 2 hours delayed in Pearl River Delta due to high proportion of scattering aerosol and negligible changed in Yangtze River Delta. The authors found that the period with the most occurrence frequency of precipitation start time is delayed and prolonged by aerosols over North China Plain, and discussed the response to precipitation to aerosol under different meteorological conditions. With the very interesting and valuable findings that include but are not limited to the parts I mention here, I believe this study is very important contribution to the science community regarding the aerosol-precipitation interaction.

We highly appreciate the reviewer's positive evaluation about our study and have made corresponding changes based on the valuable comments from the reviewer.

**Some minor comments**

Line 60: "with the increase of the aerosol" should be "with the increase of aerosol".

Thank you. We have corrected it.

Line 86-87: "in the initial stage" should be "at the initial stage", "in the development stage" should be "at the development stage".

Thank you. We have corrected them.

*Line 114-115: I would suggest adding a reference for topographic rain effect.*

We have added a reference at Lines 118-119: "Due to the topographic rain effect (Jiao and Bi, 2005), this study only selects the area with DEM less than 100 meters as the study region.".

Jiao, M. Y. and Bi, B. G.: Mesoscale structure analysis of topography-induced heavy rainfall in Beijing in summer, Meteorology, 31(6), 9-14, http://dio.org.10.3969/j.issn.1000-0526.2005.06.002, 2005, (in Chinese).

Line 134-135: I would suggest changing the description to "at a vertical interval of 125 meters".

We have changed it as suggested.

*Lines 140-141: Please provide a brief description about the method to classify the convective, stratiform, and other precipitation types.*

Following this valuable suggestion, we have provided a brief description about the method to classify precipitation types at Lines 144-166: "The method of precipitation type classification for DPR is based on different vertical motion distributions and microphysical mechanism of different precipitation types. The difference between two frequency (Ku and Ka band) observations or so-called measured dual-frequency ratio (DFRm) provides rich information to investigate the microphysical properties of precipitation. The DFRm vertical profile is controlled by the non-Rayleigh scattering effect and the path integrated attenuation difference (\deltaPIA) between two frequency channels (Le et al., 2010). The DFRm is mainly controlled by non-Rayleigh scattering effect in the ice region. Both non-Rayleigh scattering effects and  $\delta$ PIA play a role in the melting region. The DFRm is dominated by  $\delta$ PIA in the liquid precipitation region. Different precipitation types have different characteristics. As the case for convective precipitation, mixing of hydrometeors can be present in the melting layer, and in general, density of the mixture is higher than the case of stratiform precipitation (Le and Chandrasekar, 2013). Therefore, the vertical profile of DFRm has different characteristics for stratiform and convective rain according to significant on-Rayleigh scattering part and  $\delta$ PIA part. More details about the precipitation type classification method for DPR can be found in Le et al. (2010) and Le and Chandrasekar (2013).".

- Le, M., Chandrasekar, V. and Lim, S.: Microphysical retrieval from dual frequency precipitation radar board GPM, Proc. IEEE IGARSS, 3482-3485, http://dio.org.10.1109/IGARSS.2010.5652487, 2010.
- Le, M. and Chandrasekar, V.: Precipitation Type Classification Method for Dual-Frequency Precipitation Radar (DPR) Onboard the GPM, IEEE Trans. Geosci. Remote Sens., 51(3):1784-1790, http://dio.org.10.1109/TGRS.2012.2205698, 2013.

Line 174-181: The authors attempt to find suitable indicator as a proxy for CCN and they select 4-hours mean  $PM_{2.5}$  mass concentration before precipitation to investigate the impact of aerosols on precipitation. Why do not the authors choose 5-hours mean  $PM_{2.5}$  mass concentration before precipitation or the  $PM_{2.5}$  mass concentration during the precipitation to represent the CCN?

Thank you for the question. As shown in Figure R1, the correlation coefficients between the  $PM_{2.5}$  mass concentration averaged in 4 hours before precipitation and  $PM_{2.5}$  mass concentration averaged in 5 hours before precipitation are good over three study regions. However, taking diurnal variations of  $PM_{2.5}$  and aerosol accumulation effect into account, this study selects the 4-hours mean  $PM_{2.5}$  mass concentration before precipitation to investigate the impact of aerosols on precipitation.

Figure R1: The relationships between the mean  $PM_{2.5}$  mass concentration of 4 hours before precipitation ( $\mu g/m^3$ ) and the mean PM2.5 mass concentration of 5 hours before precipitation in June-August from 2015 to 2020 over North China Plain (NCP), Yangtze River Delta (YRD), and Pearl River Delta (PRD), respectively.

*Line 187: I would suggest changing the description to "The low troposphere stability (LTS) can ... " to define LTS.*

We have changed the description as suggested: We provide the full name and abbreviation of LTS when we first refer at lines 100-103: "Moreover, the changes of aerosol impacts on precipitation time with meteorological conditions that can affect precipitation have also been investigated, including the relative humidity, low troposphere stability (LTS), and vertical wind shear (WS), which are essential to aerosol-cloud-precipitation interactions (Boucher and Quaas, 2012; Fan et al., 2009; Klein, 1997; Slingo, 1987; Zhou et al., 2020).".

Line 199-200: I would suggest changing "have contributed to" to "have been used by".

We have changed it as suggested.

*Line 203: I would suggest changing "on different pressure levels" to "at different pressure levels".*

We have changed it as suggested.

Line 220-222: It seems the description here is wrong. I believe the correct description should be "Second, we rank the  $PM_{2.5}$  mass concentration observations from high to low, and define the top 1/3 of group C as polluted condition and the bottom 1/3 group C as clean condition."

We appreciate the reviewer's help figuring this out and have corrected it.

Line 240: "The diurnal variations" should be "The diurnal variation".

Corrected as suggested.

Line 250: I would suggest changing "make" to "making".

We have changed it as suggested.

Line 288: I would suggest changing the description here from "the PDFs of the precipitation duration time and when the peak time occurs after start time" to "the PDFs of the precipitation duration time and the time difference between precipitation peak and start time".

We have changed it as suggested.

Line 416: I would suggest adding "that" after "show" here.

We have changed it as suggested.

Line 531: "are" should be "is", corresponding to "response".

We appreciate the comment and have changed "response" to "responses".

*Line 590: "which are essential for improve our understanding" should be "which are essential to improve our understanding.*

We have changed it as suggested.